# Assessing Barriers Encountered by Women in Cervical Cancer Screening and Follow-Up Care in Urban Bolivia, Cochabamba

**DOI:** 10.3390/healthcare10091604

**Published:** 2022-08-24

**Authors:** Sherihane Bensemmane, Katherine Loayza Villarroel, Kevin Montaño, Elyas Louati, Carla Ascarrunz, Patricia Rodriguez, Véronique Fontaine, Samia Laokri

**Affiliations:** 1Ecole de Santé Publique, Université libre de Bruxelles (ULB), Route de Lennik, 1070 Brussels, Belgium; 2Health Services Research, Epidemiology and Public Health, Sciensano, 1050 Brussels, Belgium; 3Question Santé asbl, 1050 Brussels, Belgium; 4Instituto de Investigacion en Ciencas Sociales INCISO, Universidad Mayor de San Simon (UMSS), Cochabamba, Bolivia; 5Faculté de Pharmacie, Université libre de Bruxelles (ULB), Campus Plaine, Boulevard du Triomphe, 1050 Brussels, Belgium; 6Laboratorio de Virologia, Facultad de Medicina, UMSS, Cochabamba, Bolivia; 7School of Public Health and Tropical Medicine, Tulane University, 1440 Canal St, New Orleans, LA 70112, USA

**Keywords:** cervical cancer screening, women health, follow-up, HPV, low-income countries, health care services

## Abstract

Background: Timely detection of cervical cells infected with high-risk human papillomavirus (HPV) improves cervical cancer prevention. In Bolivia, actual screening coverage only reaches 33.3% of the target population aged between 25 and 64 years despite free cytology screening. Furthermore, 50% to 80% screened women are lost during follow-up. This study aimed at identifying factors explaining this lack of follow-up care. Method: During the first phase, face-to-face semi-structured interviews were conducted with HPV-positive women. Secondly, we explored the reasons for the non-adherence to the follow-up care: knowledge, perceptions and beliefs about HPV, as well as barriers to healthcare access, using a structured survey on Cochabamba women and healthcare professionals. Results: Barriers to effective follow-up of the targeted populations were associated with health system shortcomings, including poor service delivery at the front- and second-line, health providers shortage, inadequate training, waiting time, high direct and indirect costs of care seeking and care, complex procedures to obtain HPV screening results and poor patient–provider communication. The follow-up was perceived as extremely stressful by the participants. Conclusion: Improved communication on HPV and HPV-related cancers in terms of representation in the general population and among the health professional’s population is vital to improve access for HPV infection follow-up care.

## 1. Introduction

Cervical cancer is the fourth most common cancer among women in the world. In 2020, World Health Organization (WHO) estimated ~604,000 new cases of cervical cancer diagnosed worldwide and ~342,000 deaths attributed to cervical cancer, with about 90% of the new cases and deaths occurring in low and middle-income countries [1,2]. Almost all cervical cancers (99%) are due to persistent high-risk human papillomavirus (HR-HPV) infection [3]. However, most HPV infections are asymptomatic (or eventually associated with benign hyperproliferative lesions) and cleared by the immune system in less than 18 months. About 40 HPV types, from the alpha genus, can be sexually transmitted (with a global prevalence around 8% in sexually active population) and can infect mucosa from the anogenital tract. Based on their association with cervical cancer, these HPV are further classified as HR-HPV, such as HPV-16, 18, 31, 33, 35, 39, 45, 51, 52, 56, 58 and 59. Although twelve HR-HPV types have been identified, 70% of cervical cancers are caused by HR-HPV type 16 and 18 [1]. The HR-HPV infection is thus necessary but is not sufficient to cause cervical cancer.

According to the Pan American Health Organization (PAHO) and to the International Agency for Research on Cancer (IARC), cervical cancer is the leading cause of death for Bolivian women [4,5]. Bolivia has one of the highest rates of morbidity and mortality of cervical cancer worldwide, with a crude incidence of 34.1/100.000, an age-standardized incidence of 36.6/100.000 and a cervical cancer mortality-to-incidence ratio of 0.53 in 2020 [2,4]. The situation is dramatic, especially knowing that early prevention (vaccination) or early detection (secondary prevention) of precancerous lesions can almost completely prevent cancer development [1]. However, poverty is a factor influencing disparity in the risk of cancer development as increasing average socioeconomic levels and diminishing risk of HR-HPV persistent infection (e.g., vaccination) have been correlated with reduction in incidence rates [1].

In Bolivia, only cytological analysis, and visual inspection under acetic acid (VIAA) are included in the national screening program. However, Bolivian screening coverage is poor, being between 20–33%, and performance is not satisfactory [5,6]. New screening methods based on HR-HPV DNA detection combined with VIAA or cytological analysis can not only improve early detection of cervical cancer but can also be performed on self-collected samples [7,8]. This could provide an interesting advantage in low- and middle- income countries, allowing us to reach a larger population and therefore to increase screening coverage. A pilot study was conducted in the department of Cochabamba to enable self-sampling cervical cancer screening. During the process, we observed a low rate of follow-up in women diagnosed as HR-HPV DNA positive and/or with VIAA positive precancerous lesions [6].

Eighty percent of the Bolivian population adopt a pluralistic indigenous medicinal approach for disease treatment. This predominant approach consists of three overlapping patterns. The “family-based” pattern refers to the sickness being recognized by a family member, a relative or by the patient himself. Social networks and communities may also play a role in this first pattern, which is the primary approach leading to 60% of the cases to receive treatment. Patients also seek advice through the traditional practitioner sector (Yatiri, Naturistas, healers), which constitutes a second common pattern. Finally, patients also seek advice through medical healthcare providers [9,10].

In this study, our aim was to identify potential failures and gaps in the national cervical cancer program by exploring both the demand and supply perspectives. In the department of Cochabamba, both beneficiaries’ and providers’ perceptions were therefore analyzed around their respective experiences with the national program.

## 2. Materials and Methods

### 2.1. Study Setting and Design

Approved by the Bio-ethics committee of the Universidad Mayor de San Simon, Cochabamba, the study was performed in 2 successive phases and covered the urban and rural areas of the department of Cochabamba, Bolivia (Figure 1). Phase 1 of the study investigated the demand side through the beneficiaries’ perception, HPV-positive women screened during a pilot project [6]. The aim of the pilot was to assess acceptability and clinical effectiveness of an inexpensive HR-HPV DNA detection strategy on self-collected samples to detect high grade intraepithelial lesions and cervical cancers. This was performed in comparison with Pap smear and VIAA screening analyses. Here, the phase 1 study population also included the healthcare providers and community-based agents, “promotoras”, involved in the pilot. Semi-structured interviews were conducted in the respondent native language (Spanish or Quichua). Interviews took place face-to-face when possible or were conducted by phone (Appendix A).

In the second part of the study (phase 2), we applied structured surveys to women regardless of their HPV diagnosis status and to healthcare professionals of the department of Cochabamba (Figure 1).

The department of Cochabamba is divided into 16 provinces and 47 municipalities, and 35.9% of the population live in the municipality of Cochabamba, representing 52.6% of the urban population [11]. Interestingly, some municipalities belong to more than one province. In total, we covered 9 different municipalities grouped in 7 provinces (Figure 1 and Appendix A and Appendix A). We contacted every woman involved in the pilot project [6] as well as women from the general population in the department of Cochabamba (Figure 1).

### 2.2. Data Collection Tools and Guidelines

Phase 1 was a qualitative exploratory phase using semi-structured interviews (Appendix A). The Appendix A (semi-structured interview guides) were based on field experiences while aligned with current scientific evidence. The interviews covered participants’ knowledge around the risks of acquiring HPV infection, cervical cancer risk factors, barriers to cervical cancer screening and cervical cancer treatment. Additional information was sought to further investigate: (1) beneficiary beliefs about HPV, cervical cancer and potential impact on the care-seeking behavior and compliance pattern to treatment and follow-up care; and (2) provider perceptions around existing barriers to health care access and control services, patient–provider patterns, and common practices.

Preliminary results, obtained from phase 1, informed phase 2. In phase 2, two distinct surveys were used: one was applied to women (Appendix A), the other one to healthcare professionals (Appendix A). The survey questionnaires were hosted on the digital SurveyMonkey© platform and were sent to all HPV-positive women screened during the pilot-project, to women attending medical centers in targeted areas and to all women from the general population. There were 893,373 women (50.7%) amongst the 1,762,761 inhabitants of Cochabamba, of which 30.5% lived in rural areas [11]. An optimal sample size of 384 was therefore calculated, giving a confidence level of 95 and a margin of error of 5%.

### 2.3. Procedure and Data Collection

Data collection was conducted in 2 successive phases. In the first phase, semi-structured interviews were conducted targeting three key populations: healthcare professionals (HCP), promotoras (community-based agents) and HPV-positive patients screened during the pilot study [6]. Interviews were carried out from February 16th until July 6th 2018. Participants were also informed of the anonymity of the questionnaire and the non-binding character of all questions. They were recorded upon interviewee authorization for transcription, translation, and analysis purposes. The interviewer read the informed consent letter to the participant and obtained verbal, not written, consent prior to the interview. Interviews occurred in various places (hospital/health center waiting room or home) or were done remotely (phone).

Data collection of phase 2 occurred between 9–27 July 2018. Both questionnaires were used simultaneously, one targeting the supply side (health care personal) and the other the demand-side (women from the general population, attending medical centers). Paper version of the HCP questionnaire was distributed in hospitals and community centers, and a survey link was shared to augment the scope. To reach the general population, the women’s questionnaire was promoted through social media (e.g., Facebook and WhatsApp) and direct contacts (e-mails). To increase study representativeness, reduce digital illiteracy bias and reach a broader audience, a paper version of the questionnaire was also made available and distributed in selected places (e.g., hospitals, health centers, public/community places such as cafés). When approaching potential respondents, we offered the option to read the question with them to include all women despite potential language or literacy barriers, as 8.3% of 15 years old or older women are illiterate (4.8% in urban areas, and 16.9% in rural areas) [11]. All women over 15 years and living in Cochabamba were invited to participate. Indeed, in Bolivia, the law (from 3 July 2012) established that women of 18 years and more should receive a Papanicolaou test, and national statistical studies on sexual and reproductive health are targeting women over 15 years. Bolivian women are at high risk of cervical cancer, they claimed a median age of 17 at first sexual intercourse and seldom use condoms [12]. Therefore, although the previous pilot project on cervical cancer screening was performed on women with 25 years or more, here, in this survey assessing, among others, the level of knowledge on HPV and cervical cancer, we wanted to have a picture of the population at risk, even in the future, because they are also those having to decide for their vaccination against HR-HPV or not.

Interviewers provided an informed consent letter to the participants and obtained verbal consent prior to the interview (phases 1 and 2). Surveyed patients had to opt in to answer the online version of the questionnaires (phase 2).

### 2.4. Data Analysis

Transcripts of qualitative interviews (phase 1) were explored to identify recurrent themes and were triangulated by analysts and discussed with interviewers to ensure accurate transcription. The coding revealed a series of themes: (1) health provider and women’s beliefs, (2) health providers and women’s knowledge, (3) factors influencing health seeking behaviors, (4) the possibility to work with traditional healers and churches and, finally, (5) barriers. Representative quotes were selected to showcase the results from the interviews. Participants were anonymized (Appendix A).

Descriptive statistics were used to analyze data extracted from the surveys (phase 2). Logistic regression, using R software, was performed to investigate the role of age and education level in HPV-related knowledge. Multiple logistic regression models included covariates such as age, level of education, and medical care settings chosen by women for general and gynecological check-ups. We checked the interaction between age and education and used backward elimination as an effect-selection method. We used multiple regression to obtain odds ratio (OR), Wald statistics and Wald 95% confidence interval (CI). We performed a Chi-squared test to investigate the link between HCP characteristics (training, medical specialty) and choice of treatment, verification of HPV test results or the level of explanation provided to patients. Sample calculation was performed using Andrew Fisher’s Formula.

## 3. Results

In phase 1, we conducted semi-structured face-to-face or phone interviews with 10 health professionals involved in the HPV self-sampling project (Appendix A), with 15 HR-HPV positive women (Appendix A) and with four community women working as “promotoras” in the pilot project (Appendix A). In phase 2, there were 244 respondents for the survey questionnaire targeting women and 36 respondents for the healthcare professional questionnaire.

### 3.1. Preliminary Interviews to Identify Common Patterns Possibly Involved in Reduced Cervical Cancer Screening and Follow-Up Care (Phase 1 Study)

#### 3.1.1. Participant Characteristics

Among the 10 healthcare professionals interviewed in phase 1, only one interviewee was male. HCP were represented mainly by gynecologists (50%) and nurses (40%). The median age of interviewees was 39.5 (Q1 37; Q3 43.8). Four “promotoras” community women were interviewed (median age 38.5; Q1 26.5; Q3 51.3).

Among the 15 women that screened HR-HPV positive through self-collected samples, one completed both questionnaires (for healthcare providers and patients). HPV screened women were between 27 and 60 years old (data missing for two women, median age 35 (Q1 30; Q3 37), three had no employment, and four were merchants).

#### 3.1.2. HPV Related Knowledge

The lack of knowledge was often mentioned in the interviews by the healthcare providers as a potential cause of the low adherence to follow-up care. Sixty percent of HCP considered patients having misconception, no or low level of knowledge on HPV (Table 1). To illustrate this point of view of HCP, we chose the following quote:


*“I believe that women don’t come to follow-up mainly because of lack of knowledge, lack of culture, but also because of the cost of the treatment because it is not completely free. They may give good drugs, but they are expensive and therefore not accessible.”*


Patients acknowledged their lack of understanding, nonetheless they point out the fact they did not receive sufficient explanations or information:


*“What scares me is that sometimes they don’t explain you well. It came to a point where I did not understand anything, and I thought the delivery of my results was my death sentence. When I did the Papanicolaou in Tiquipaya (not in the project) the doctors told me that I had lesions and that I had to follow a treatment but did not tell me why I had to follow this treatment.”*
Woman #3

Patients with prior knowledge or access to knowledge had misconceptions or incomplete information:


*“HPV is a sexually transmitted virus. I don’t know what the risks of infection are. I got this information from my sister who is a nurse.”*
Woman #5

Knowledge was often mentioned as an issue, as well as information given to patients. Furthermore, promotoras, who were community agents facilitating access to knowledge and raising awareness, recommended improving their training:


*“I would like the explanations to be given in Quechua because here in the countryside people speak Quechua. I would also like them to explain things to us using a mannequin to be able to visualize the explanations that the doctors give.”*
Promotora #1

Overall, 40% of HCP mentioned cultural differences and 20% mentioned level of education as explanations for the lack of screening or follow-up, while 53% of women considered the information on testing or treatment they received from HCP or promotoras (how, when, and where) to be insufficient.

#### 3.1.3. Traditional Practitioners

According to some healthcare professionals, women preferred to refer to traditional healers for treatment, because of a lack of trust in the health care system and malpractices:


*“In any case, patients do not have much confidence in what the health center does, and they prefer to be tested in a private institution.”*
HCP #9

However, this was not found in the discourse of all women and was not perceived as incompatible:


*“I would not agree that the doctor works with a traditional healer. I do not think the church, or my faith can cure diseases. I think there should be more medical attention.”*
Woman #1, woman screened within the pilot project

#### 3.1.4. Treatment, Mistrust and Care Practices

Promotoras informed patients that if the result of the self-sampling test is positive, they will have to go to the hospital and may have a second test.

Seven HCP said they will choose a hysterectomy as treatment, although sufficient clinical indication is not always available:


*“Some patients coming for screening already had a hysterectomy following Papanicolaou results. Doctors don’t have the right information to perform hysterectomies.”*
HCP #1

This is confirmed by other HCP:

*“In patients over 50, we perform hysterectomy because they no longer need it [the uterus].”* HCP #9. Another HCP revealed: *“the results showed that she had initial cancer, she was given a hysterectomy and now she is fine.”*HCP #8

From the perspective of the patient, difficulties in understanding the HCP take-home message and human resources issues (of all kinds) could contribute to the lack of follow-up care:


*“The test result said: highly at risk or something like that. I was told to go that day, but I could not. I then went another day, but the staff was no longer there.”*
Woman #11

The patient–physician relation was not collaborative, exacerbating mistrust expressed by some interviewed women and tales that other women had heard. Seventy-five percent of promotoras mentioned fear of death, hysterectomy or cancer as a notion shared by many women and themselves prior to their training. In some cases, as mentioned by HCP, the proposed treatment could be disproportionate:


*“I’m 27 years old. I did the HPV self-sampling test, it was positive. I was sent to Cochabamba Hospital and the doctor did an operation to remove my womb (uterus). I did not want to do the hysterectomy (…) The infection was low grade. The doctor told me that there were worms in my uterus and that it was going to cause me cancer.”*
Woman #15

#### 3.1.5. Barriers to Access Health Care

A deficit in trained healthcare professionals as well as accessibility to appropriate centers were identified as barriers to access health care:


*“In our health center, we do not have a specialized gynecologist, the people who take care of the women are general practitioners and they are the ones who do the tests.”*
HCP #6 at Pucarita

A delay in accessing results can be multifactorial, although within the pilot project results were available within 15 days at laboratories, delays were encountered mainly because of delivery, loss, or organization of collection slots:


*“The lab had the results in two weeks, but we could only get them once a month. Sometimes the results were given in a month and sometimes it even took longer.”*
HCP #2

Cost has been identified as a barrier by healthcare professional as well as by female patients. During the pilot project, screening was available and promoted by the presence of health professionals and promotoras in rural areas, and the screening could be done by self-sampling. However, there was an important drop-out in treatment and follow-up care. Several potential causes were identified: direct cost of the treatment or indirect costs, such as transportation, or perspective of losing one day of salary, especially for informal workers (e.g., merchants), impeding their access to follow-up care:


*“The money, being a widow with children, and the distance from the hospital where the free follow-up was done kept this woman from going for the follow-up.”*
Promotora #1


*“Treatment is free until it is an invasive cancer. After that, the patient must pay. The treatment is very expensive and not included in the free insurance because the government cannot pay.”*
HCP #9

Furthermore, other challenges were identified by interviewees such as delays in the result delivery, understaffing, etc. These explanations received by HCP and promotoras can explain dropouts:


*“I did the self-sampling at Villa-Tunari, I received the results. I was thinking of going to the follow up, but I don’t know where it is.”*
Woman #7

### 3.2. Surveys to Verify Common Patterns Possibly Involved in Reduced Cervical Cancer Screening and Follow-Up Care (Phase 2 Study)

#### 3.2.1. Participant Characteristics

Among the 244 surveyed women, 187 (76.6%) knew about HPV, 52 were screened for HPV (21.3%), and among them, five never received their results. Fifteen women who had undergone screening were HPV-positive (30.6%), and eight had uncertain HPV-status, of which one needed to be retested (Figure 2). Among the surveyed women, only two were involved in the pilot project [6], but we do not know whether they were interviewed during phase 1 or not.

Overall, 21.3% were aged between 15 and 20 years, 55.7% were aged between 21 and 35 years, 17.6% were aged between 36 and 50 years, and women above 50 only represented 5.3% of our population sample. The majority followed a university education (55.7%), 31.1% completed secondary or university baccalaureate (Bachelor), 7.4% had a primary education, 1.6% never had any training at any level, 2.9% did not finish their study and 1.2% had a technical training.

Among the 36 surveyed HCP, general practitioners represented 38.9% of the sample population, health care assistants represented 22.2%, gynecologists 16.7% and nurses 2.8%.

#### 3.2.2. HPV Related Knowledge

Overall, 15.6% participants had never been aware of HPV and 18% never had a gynecological follow-up of any kind (Appendix A). Among the women who had already heard of HPV, only 16.51% received information about HPV from healthcare professionals (6.8% from a general practitioner and 9.71% from gynecologists) and 5.83% received it through promotoras. Most of the time, information was transmitted otherwise (19.9% via a close person, 40.8% via communication such as television and 17% sought by themselves). The link between age and HPV knowledge was not significant (Wald statistic 0.85, *p* = 0.84); however, level of education impacted the odds of having prior knowledge of HPV (Table 2). Education was not a predictor of the odds of being screened for HPV (Wald statistic 7.42, *p* = 0.28).

Overall, 75% of healthcare professionals said they have received specific training on infection and detection of papillomavirus and 97.2% have received training on cervical cancer. There was no correlation between HCP HPV training and the level of explanation provided to patients (χ^2^ = 4.3, *p*-value = 0.2).

#### 3.2.3. Traditional Practitioners

Among the 244 female respondents (regardless of HPV knowledge), only 0.4% used traditional healers or practitioners for general consultation and none did gynecology check-ups.

However, 14 HCP, 38.9% of our sample population, argued that women would rather go to traditional practitioners or use prayers to heal themselves instead of seeking help among healthcare providers. Overall, 41.7% (15/36) healthcare professionals responded that they were willing to collaborate with traditional practitioners, while 13 (36.1%) would refuse to collaborate with traditional practitioners. Only one healthcare professional was already working with traditional practitioners within their practice.

#### 3.2.4. Treatment, Mistrust and Care Practices

Overall, 90.91% of providers (N = 33) used the Papanicolaou test as the first choice for screening. Amongst the 49 HPV-tested women in our sample population, 59.18% received the Papanicolaou test and 14.29% used HR-HPV self-screening facilities. Although the majority did have access to training, only 58.3% of healthcare professionals reported verifying screening results and 63.9% (23/36) checked the HPV status and lesion grade before proposing a treatment. There was no correlation between the HCP specialty and the verification of the HPV screening results (χ^2^ = 3.8, *p*-value = 0.9) or the lesion status when choosing a treatment (χ^2^ = 5.8, *p*-value = 0.2). Moreover, the three most recommended treatments for benign lesions (Table 3) were drug treatments (41.7%), cryotherapy (27.8%) and conization (27.8%). Hysterectomy was mostly proposed when treating invasive lesions (61.1%).

Among HPV-positive women (N = 15), 20% received cryotherapy, 20% medication, 6.7% had a hysterectomy and 33.3% received no treatment, the rest were not certain or had to repeat the tests. Among women who responded “uncertain” regarding their HPV status (N = 8), 25% received cryotherapy and 20% medication. Overall, 26.7% were followed in a private hospital, 20% in a public hospital, 2 (13.3%) were followed by the project gynecologist and one woman was followed by a private doctor.

#### 3.2.5. Barriers to Access Health Care

Healthcare professionals considered that the husband of the family could prevent the patient attending the screening program (27.8%) or to follow a treatment or a consecutive follow-up (19.4%) (Table 4). However, only 0.5% women considered their husband or family as a barrier to get screened, and none considered it could be a barrier to receive treatment or follow-up care.

Communication is an important barrier. Overall, 26.7% of women under HPV treatment or follow-up considered that healthcare professionals did not give them enough information to understand their situation. Thirty out of thirty-six (83.3%) healthcare professionals say they provided information, but only 60% of them (18/30) checked the level of patient understanding. Three HCP respondents feel they were not concerned by the need to inform patients, given their role within their institution. Three main ways are used to share results with patients (Appendix A in Supplementary material File S1): phone calls, general practitioners and gynecologists. However, 50% of HCP opted to wait for a consultation with the patient.

Interestingly, 26.7% of surveyed women consider the waiting time as a barrier (Table 5). Healthcare professionals mentioned the lack of material (19.4%) or qualified staff (2.8%), and cost of the test (8.3%) as a barrier to follow-up care (Table 4). Additionally, 19.8% of women and 25% of healthcare professionals think that the cost of HPV screening is a barrier (Table 4 and Table 5).

## 4. Discussion

We aimed to identify barriers among Cochabamba’s women, including determining their knowledge about cervical cancer risk factors, identifying cultural beliefs and cultural health behaviors that could impede cervical cancer screening and treatment.

Our study suggests that, in Cochabamba, knowledge about cervical cancer and HPV was extremely low even among highly educated women and among health providers. Women screened using self-collected samples from the pilot project (phase 1) still had unclear or wrong information about HPV and cervical cancer.

A study to assess the communication difficulties in sexual healthcare should be conducted by surveying caregivers and patients to better understand the difficulties encountered by each other in order to reduce this communication gap.

The care provider sex and the reaction of the husband’s perceived role on the care seeking decision for screening were rarely raised by women as potential barriers to health care (1%). However, care providers attributed the lack of adherence to treatment and follow-up to care user spiritual and cultural beliefs, their refusal to be treated by male professionals (in phase 1: 25% of promotoras, N = 4; 20% of HCP, N = 10), the presumed need of the husband’s permission (in phase 2: between 20 and 28%) and their beliefs that women would generally consult traditional practitioners such as Yatiris/Naturistas (phase 1: 60% of HCP, N = 10; phase 2: 39% of HCP, N = 36). This was not confirmed when analyzing women’s survey responses. However, our survey did not include most rural areas surrounding Cochabamba. Indeed, although some health centers had patients from rural areas, we did not capture all remote areas (Figure 1 and Appendix A). The HCP perception could be linked to the notion of culturalism, described by Alexandra Nacu [13]. Indeed, HCP could neglect the potential institutional issues or factors related to themselves and could instead perceive women’s (cultural) characteristics as major factors explaining low women’s follow-up care [13].

The lack of follow-up, as observed in our pilot study, eventually due to the various barriers highlighted in this study, emphasized the requirement for a see and treat screening strategy in this population. Indeed, this could limit the loss of patient follow-up, as previously observed in various low-income countries [14,15]. However, VIAA and rapid HR-HPV detection technics should be performed to reduce the risk of overtreatment [16,17].

In a Bolivian rural area, in Roboré town in the province of Chiquitos, a study showed low screening coverage (41–46%) and low opportunity for treatment (13–16.7%) [18]. In agreement with our results, the lack of infrastructure, the scarcity of HCP involved in cervical cancer and the low knowledge about cervical risk factors were identified as potential factors hindering cervical cancer screening and treatment [18]. In this study, some HCP chose hysterectomy and medication as potential treatments for a benign cervical lesion, although only cryotherapy, thermoregulation and excision (among others by loop electrosurgical excision procedure, LEEP) are recommended (among others by WHO) and hysterectomy is only recommended in the case of invasive cancer [19,20,21]. This emphasized the need for adequate and actualized clinical training for all healthcare providers.

Moreover, financial barriers were also mentioned by women in both study phases. Direct cost (screening, medical consultation, and treatment fees) was only covered for women who were included in the pilot project, but indirect health costs, such as transport fees, should further be included. It is worth noting that the 2012 law 252 ensures that 18 or older workers have access to one day off per year for HPV testing [22], and the 2013 law guarantees free Papanicolaou cytology testing [23,24]. However, according to the last official census [11], only 39% of women were employed or actively seeking employment. Subsequently, female students, women in informal labor or housewives would not be able to enjoy disposition of the 2012 law 252. Moreover, only the screening using the Papanicolaou test is free, consultation and indirect costs are thus out-of-pocket expenditures. These could be considered as barriers in general HPV-related care (Table 1 and Table 5).

All sampled HCP worked in public hospital; however, surveyed women were consulted in all settings (phase 2). Their gynecological consultations occurred in public hospitals (26.2%), in private hospitals (27.1%), in health centers (14.8%), and notably 18% had no check-up whatsoever. In phase 1, all women had HPV screening and follow-up care, carried out in public settings. Our results show structural barriers (equipment, human resources), regardless of setting type.

This study shows the need to strengthen health care services (prevention, testing, and follow-up care), to train HCP and to increase HPV program awareness. In phase 1 of our study, we recruited women having received an HR-HPV positive result from their self-collected biosamples. We could not rely on random sampling methods to select the study participants. However, this study was conducted in different regions of Cochabamba, rural and urban ones. We had highly educated participants, but also participants with a low level of education. In the second phase, we surveyed women attending the same medical centers and broadened our sample population to all women, whether they were screened or not. We confirmed identified barriers to health service access and factors involved in the dropping out of follow-up. However, the number of respondents was limited. Therefore, although informative, their responses cannot be considered representative of the area of Cochabamba.

## 5. Conclusions

Health system sub-performances and direct (for invasive cancer treatment) and indirect (such as transport fee) economic factors were the main reported key factors explaining the lack of follow-up and the low uptake in health care services in the area of Cochabamba. In the future, increasing financial support in Bolivian clinical health centers to increase human and technical resources (especially in see and treat strategy) could only partly reduce provider and healthcare system-level barriers, by reducing, for example, long waiting periods before healthcare. Additionally, our results emphasized the need of implementing quality improvement interventions, to train health providers not only in recent technical clinical recommendations (to improve treatment choice for example), but mainly in health care communication. Simultaneously, popularized scientific information explaining the relative risk of sexually transmitted HPV infections in the development of the cervical cancer could be of high value, not only to increase cervical cancer awareness, but also to prevent unnecessary anxiety. Administrative tasks, still performed manually, are time-consuming to the detriment of the availability of health personnel who are already understaffed and overloaded. Digitalization could also potentially improve this issue.

## Figures and Tables

**Figure 1 healthcare-10-01604-f001:**
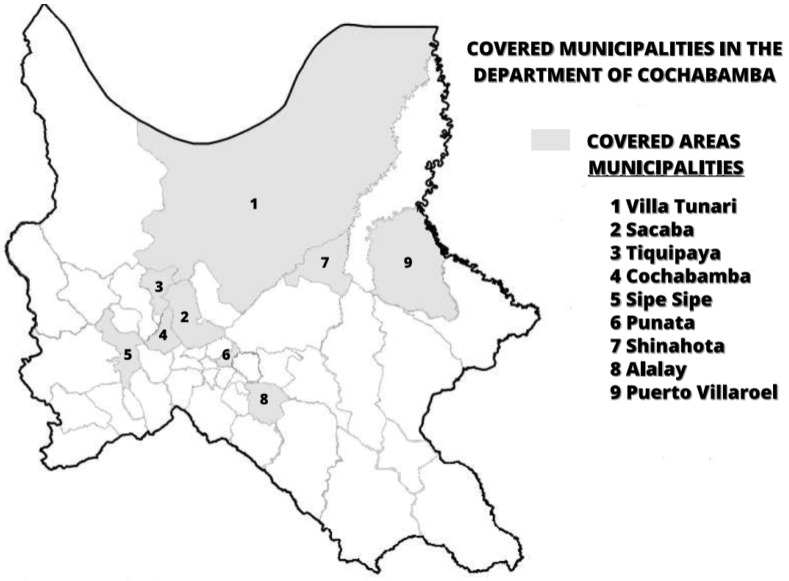
Map showing study covered municipalities in the department of Cochabamba (phases 1 and 2).

**Figure 2 healthcare-10-01604-f002:**
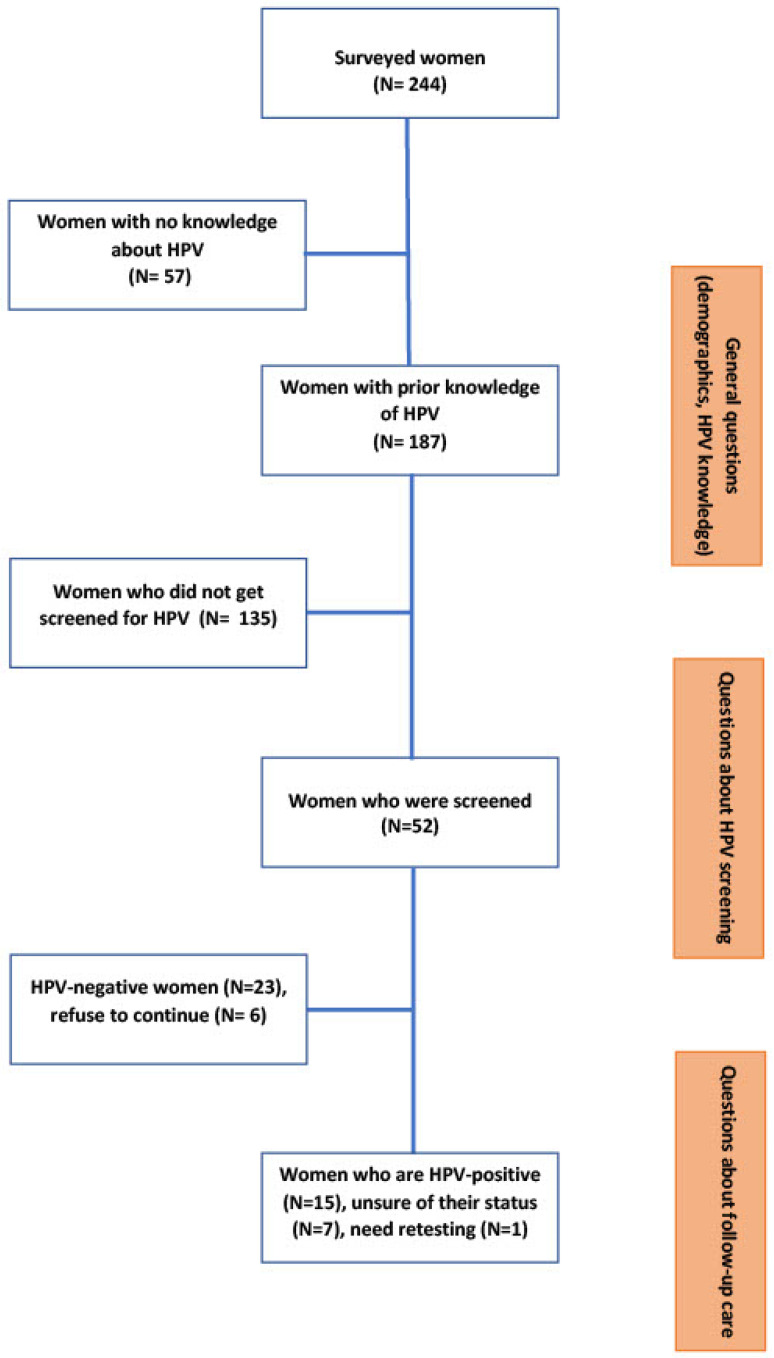
Flowchart describing surveyed women’s questionnaire sections (phase 2).

**Table 1 healthcare-10-01604-t001:** Percentage of interviewees in phase 1 mentioning specific response in regard to a specific category of cervical cancer healthcare characteristic.

Categories		Women N = 15 (%)	Promotoras N = 4 (%)	HCP N = 10 (%)
Applied treatment	Hysterectomy	1 (6.67)	0	7 (70)
	Cryotherapy	0.00	0	1 (10)
	Conization	3 (20.00)	0	1 (10)
	Medication	0.00	1 (25)	0
	Drop-out	6 (40.00 *)	0	0
	Traditional healers	1 (6.67)	0	0
Women’s HPV knowledge	No prior knowledge or low level	4 (26.67)	2 (50)	3 (30)
	Beliefs and misconception (HIV, HPV = cancer, disease, infidelity)	0.00	2 (50)	3 (30)
	Prior knowledge	2 (13.33)	1 (25)	0
	Good level	1 (6.67)	0	0
Traditional practitioners	Never used/Not in favor	3 (20.00)	0	2 (20)
	Not frequent/Conditional use	3 (20.00)	4 (100)	2 (20)
	Frequent use	0.00	0	6 (60)
	Open to collaboration between tradi-practitioners and HCP	2 (13.33)	1 (25)	3 (30)
Barriers for testing and treatment/follow-up	Cost of the screening/testing	0.00	2 (50)	1 (10)
	Cost of treatments	1 (6.67)	0	3 (30)
	Indirect costs (e.g., transports)	0.00	1 (25)	0
	Delay to receive results	6 (40.00)	0	3 (70)
	Lack of (trained) staff	1 (6.67)	0	3 (30)
	Lack of material	0.00	0	1 (10)
	Lack of training	0.00	0	0
	Distance to centers	2 (13.33)	2 (50)	2 (20)
	Waiting time/Loss of a paid working day	2 (13.33)	3 (75)	2 (20)
	The HCP/promotora explanation was insufficient	8 (53.33)	3 (75)	3 (30)
	Difficulty to access information	0.00	2 (50)	0
	Husband or family do not allow patient to come	0.00	1 (25)	5 (50)
	The difficulty of freeing oneself from family obligations	2 (13.33)	2 (50)	2 (20)
	The sex of the doctor	0.00	1 (25)	2 (20)
	Healthcare professional attitude/discrimination	0.00	1 (25)	0
	Cultural or religious barriers	0.00	0	4 (40)
	Level of education	0.00	0	2 (20)
	Embarrassment	3 (20.00)	2 (50)	1 (10)
	Fear of results/of having a cancer/of getting hysterectomy	4 (26.67)	3 (75)	6 (60)
	Distrust	0	1 (25)	3 (30)
	No barriers	0	0	0

* We did not take into account Xiomara, who moved to Chili and was seeking treatment there.

**Table 2 healthcare-10-01604-t002:** Logistic regression results (phase 2).

N = 242	Predictor	Prior Knowledge of HPV		Being HPV Screened	
		OR	95% CI	OR	95% CI
Age	15–20 years old	NA	-	0.19 **	0.5–0.64
	21–35 years old (Ref.)	NA	-	1	-
	36–50 years old	NA	-	1.3	0.61–2.77
	Over 50 years	NA	-	0.55	0.12–2.59
Education	No education	11.4 *	1.47–88.44	NA	-
	Uncompleted primary education	0.51	0.01–30.2	NA	-
	Uncompleted secondary education	11.4 *	1.47–88.44	NA	-
	Primary education	9.14 ***	3.3–27.59	NA	-
	Secondary education	2.86 *	1.27–6.42	NA	-
	University education	1	-	NA	-

NA: Not applicable means the logistic model did not retain the variable as predictor and did not compute OR. Significant codes, *p* values: * *p* < 0.05, ***p* < 0.01, *** *p* < 0.001.

**Table 3 healthcare-10-01604-t003:** Treatment that HCP would propose according to lesion grade (phase 2).

Type of Treatment *	Benign Cervical Lesions N = 36 (%)	Invasive Cervical Lesions N = 36 (%)	Uncertain Test Results N = 36 (%)
Hysterectomy	1 (2.78)	22 (61.11)	2 (5.56)
Cryotherapy	10 (27.78)	5 (13.89)	1 (2.78)
Conization	10 (27.78)	6 (16.67)	1 (2.78)
Medication	15 (41.67)	3 (8.33)	4 (11.11)
Referral to a specialist/gynecologist	7 (19.45)	9 (25)	5 (13.89)
Radiotherapy	0	1 (2.78)	0
Retesting	0	1 (2.78)	11 (30.56)
None	2 (5.56)	1 (2.78)	9 (25)

* Respondents could select multiple choices.

**Table 4 healthcare-10-01604-t004:** Barriers perceived by surveyed HCP (phase 2).

Reasons *	Screening N = 36 (%)	Follow-Up N = 36 (%)
Cost of the test or treatment	9 (25)	3 (8.33)
Lack of material or medications	7 (19.44)	6 (16.67)
Lack of training to perform the test	2 (5.56)	4 (11.11)
Work overload	2 (5.56)	2 (5.56)
Lack of professionals	2 (5.56)	1 (2.78)
Husband or family do not allow the patient to come	10 (27.78)	7 (19.44)
Women do not accept a male physician performing the Papanicolaou	1 (2.78)	0
Issues retrieving results	3 (8.33)	3 (8.33)
No barriers	8 (22.22)	11 (33.33)

* Respondents could select multiple choices.

**Table 5 healthcare-10-01604-t005:** Barriers perceived by surveyed women (phase 2).

Reasons *	Screening N = 187 (%)	Follow-Up N = 15 * (%)
Cost of test/treatment	37 (19.79)	2 (13.33)
Cost of transportation	6 (3.21)	0
Distance	15 (8.02)	4 (26.67)
Healthcare professional attitude	37 (19.79)	1 (6.67)
The HCP explanation was insufficient	46 (24.6)	4 (26.67)
Husband or family do not allow the patient to come	1 (0.53)	0
Waiting time	50 (26.74)	3 (20)
Difficulties to obtain a day-off from work	27 (14.44)	4 (26.67)
The difficulty of freeing oneself from one’s family obligations	13 (6.95)	1 (6.67)
No barriers	53 (28.34)	3 (20)

* Respondents could select multiple choices.

## Data Availability

Authors can be contacted and can share information upon request. Data is available at DOI: 10.5281/zenodo.7014841.

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
