# Peer review of "Assessing Barriers Encountered by Women in Cervical Cancer Screening and Follow-Up Care in Urban Bolivia, Cochabamba"

_healthcare, 2022, doi:10.3390/healthcare10091604_

Round 1

Reviewer 1 Report

This paper is a mixed-methods study examining barriers of cervical cancer screening and follow-up care in Bolivia. The study question is important in Bolivia, given the extremely low screening rates. The qualitative component of this study is very informative however, the quantitative portion can be strengthened with additional statistical tests and modeling, rather than purely reporting descriptive univariate analyses. The authors also note that the results are generalizable, however, the convenience sampling and small sample size suggests that the results may be highly variable with limited generalizability. Further, I believe there should be fairly extensive editing of English language and writing style, as well as reorganization of some of the paragraphs to facilitate a better flow.

Abstract

The abstract is generally well written.

Introduction

1.     Line 37: change “first” to “leading”.

2.     To give more background of the cervical cancer burden in Bolivia, I suggest adding incidence and mortality rates of cervical cancer rates in Bolivia.

3.     Lines 51-52: I don’t understand this wording.

4.     I suggest to start the introduction with the broader description of cervical cancer burden worldwide and then go into details about Bolivia.

5.     Line 53: change “incriminated” to “identified”.

6.     Line 66: what is at 80%? This is unclear.

7.     Line 69: What do the authors mean by 60% of a treatment decision?

8.     Lines 73: Is there a second aim of this study? If not, please remove the word “firstly”.

Methods

1.     Please add a brief summary of the details of the pilot study in the methods.

2.     Is it possible to provide English-translated versions of Appendices 1-3?

3.     Please add details on the process for how the women in phase 2 were recruited, selected, and invited.

4.     Although statements for IRB approval and informed consent were written in sections at the end of the manuscript, please add a statement on these in the methods.

5.     Tables S2-S4 shows false identities of participants, but this can easily be misleading. I suggest removing the subjects’ false names and use another form of identification (e.g., participant #).

6.     Line 87: what does it mean “women who had or not HPV”? Do you mean women, regardless of HPV status, were administered a structured survey?

7.     Line 100: change “fed” to “administered”.

8.     Line 102: Please clarify whether the questions about cancer risk factors were about cervical cancer risk factors.

9.     Did the authors adjust for any confounding variables in the regression analyses? Were the ORs just crude ORs, examining the association of age on HPV knowledge? And another crude OR examining education level on HPV knowledge? If so, these estimates may be biased. Please add more details regarding this analysis.

10.  The analysis would be strengthened by adding more statistical tests and modeling, rather than basing the conclusions solely on percentages from small sample sizes.

Results

1.     Line 159: I don’t think we can say 50% is the majority.

2.     Line 200: Pap tests should not be performed annually. Lucy’s (HCP) quote should not be highlighted, as this is misleading information coming from a provider.

3.     Table 1: Hysterectomy is spelled incorrectly.

4.     Where are the logistic regression results shown in a table? I only see the results written in the text, with no supporting table.

5.     All tables would benefit by adding n and %s, instead of only the %s.

6.     Line 314: what types of medication are being recommended by HCPs for benign cervical lesions? I do not think there is a medication for this. Also, why are HCPs recommending hysterectomy for a benign cervical lesion? There seems to be a lot of lack of education in HCPs.

Discussion

1.     Given the identified barriers, I suggest the authors add discussion on the importance of implementing quality improvement interventions among providers to reduce provider- and system-level barriers.

2.     I do not think this study is generalizable, as the authors stated. The sample was not randomly selected and the sample size is too small, warranting high variability. We cannot make an overarching conclusion based off of such small sample sizes, especially if characteristics vary between each person.

Author Response

Reviewer 1 :

This paper is a mixed-methods study examining barriers of cervical cancer screening and follow-up care in Bolivia. The study question is important in Bolivia, given the extremely low screening rates. The qualitative component of this study is very informative however, the quantitative portion can be strengthened with additional statistical tests and modeling, rather than purely reporting descriptive univariate analyses. The authors also note that the results are generalizable, however, the convenience sampling and small sample size suggests that the results may be highly variable with limited generalizability. Further, I believe there should be fairly extensive editing of English language and writing style, as well as reorganization of some of the paragraphs to facilitate a better flow.

We thank the reviewer for its comments that we have taken in consideration to improve our manuscript. As explained hereunder we performed additional statistical analyses, we improved the manuscript writing style and we reorganized some paragraphs for clearer take-home messages.

Abstract

The abstract is generally well written.

Introduction

  1. Line 37: change “first” to “leading”.

Thank you for the suggestion, this change is done.

  1. To give more background of the cervical cancer burden in Bolivia, I suggest adding incidence and mortality rates of cervical cancer rates in Bolivia.

We agree with this comment. This is now done lines 61-63 in the pdf file.

  1. Lines 51-52: I don’t understand this wording. The introduction has been reorganized, this sentence has been rewritten.
  2. I suggest to start the introduction with the broader description of cervical cancer burden worldwide and then go into details about Bolivia. This is now done.
  3. Line 53: change “incriminated” to “identified”. Done.
  4. Line 66: what is at 80%? This is unclear. This is now better explained, line 80 in the pdf file
  5. Line 69: What do the authors mean by 60% of a treatment decision? This is now better explained, line 84 in the pdf file
  6. Lines 73: Is there a second aim of this study? If not, please remove the word “firstly”. Indeed, removed.

Methods

  1. Please add a brief summary of the details of the pilot study in the methods. Done, lines 97-101 in the pdf file.
  2. Is it possible to provide English-translated versions of Appendices 1-3? We prefer to let the original versions in case it could be of use for other studies in South America.
  3. Please add details on the process for how the women in phase 2 were recruited, selected, and invited. The details are provided in the “Methods” section under the title “Procedure and data collection” lines 225-237 in the pdf file. Some additional information have been added.
  4. Although statements for IRB approval and informed consent were written in sections at the end of the manuscript, please add a statement on these in the methods.

This is now done in the “Methods”, in the” study setting and design” and “procedure and data collection” sections.

  1. Tables S2-S4 shows false identities of participants, but this can easily be misleading. I suggest removing the subjects’ false names and use another form of identification (e.g., participant #).

This is now changed everywhere.

  1. Line 87: what does it mean “women who had or not HPV”? Do you mean women, regardless of HPV status, were administered a structured survey?

Thank you for this suggestion. This is now corrected (line 163 in the pdf file).

  1. Line 100: change “fed” to “administered”. Rephrased.
  2. Line 102: Please clarify whether the questions about cancer risk factors were about cervical cancer risk factors. This is now clarified line 179 in the pdf file.
  3. Did the authors adjust for any confounding variables in the regression analyses? Were the ORs just crude ORs, examining the association of age on HPV knowledge? And another crude OR examining education level on HPV knowledge? If so, these estimates may be biased. Please add more details regarding this analysis.

More details about our statistical analyses are now lines 262-277 in the pdf file.

  1. The analysis would be strengthened by adding more statistical tests and modeling, rather than basing the conclusions solely on percentages from small sample sizes.

This is now done. See in Methods lines 262-277 in the pdf file but also in the Results lines 517-519 and lines 522-523, lines 562-577 and lines 555-557 in the pdf file and Table 2.

Results

  1. Line 159: I don’t think we can say 50% is the majority.

Indeed, we rephrased it line 290 in the pdf file.

  1. Line 200: Pap tests should not be performed annually. Lucy’s (HCP) quote should not be highlighted, as this is misleading information coming from a provider.

This is now removed to avoid misinformation.

  1. 3. Table 1: Hysterectomy is spelled incorrectly.

We corrected.

  1. Where are the logistic regression results shown in a table? I only see the results written in the text, with no supporting table.

Additional statistic analyses are now provided: see in Methods lines 262-277 in the pdf file but also in the Results lines 517-519 and lines 522-523, lines 562-577 and lines 555-557 in the pdf file and Table 2

  1. All tables would benefit by adding n and %s, instead of only the %s. This is now done in all tables.
  2. Line 314: what types of medication are being recommended by HCPs for benign cervical lesions? I do not think there is a medication for this. Also, why are HCPs recommending hysterectomy for a benign cervical lesion? There seems to be a lot of lack of education in HCPs.

We asked these questions to assess the level of knowledge of HCPs. Some sentences and references (18, 19 and 20) for recommendations have been added in the discussion lines 668-673 and in the conclusion line 771.

Discussion

  1. Given the identified barriers, I suggest the authors add discussion on the importance of implementing quality improvement interventions among providers to reduce provider- and system-level barriers.

This is now done 630-632, 672-673 and in the conclusions.

  1. I do not think this study is generalizable, as the authors stated. The sample was not randomly selected and the sample size is too small, warranting high variability. We cannot make an overarching conclusion based off of such small sample sizes, especially if characteristics vary between each person.

This is now deleted line 700.

Reviewer 2 Report

It was a pleasure reading the paper “Assessing barriers encountered by women in cervical cancer screening and follow-up care in urban Bolivia, Cochabamba”. I believe it addresses an important topic and has the potential to strengthen the area of literature in barriers to HPV-related care, specifically in Bolivia. The findings have implications for future interventions that could strengthen provider training and cultural sensitivity, as well as patient knowledge and follow-up care. I have included specific recommendations below to strengthen this paper.

Abstract:

Lines 16-17: “Timely detection of cervical cells infected with high-risk human papillomavirus (HPV) improves cervical cancer screening.” I think a clearer statement here is that timely detection improves cervical cancer outcomes or cervical cancer prevention, not screening itself.

Line 17: “…actual screening coverage only reach…” should be “…actual screening coverage only reaches…”

Introduction:

Line 48: The acronym WHO should be defined here as it is the first use.

Line 51: What do the authors mean by “Those”? Are you referring to HR-HPV infections? If so, I would state that to make this sentence clearer. This entire sentence can also be rephrased as it is unclear as written.

Line 57: “performance are not satisfactory” should be “performance is not satisfactory”

Line 58: “combined to” should be “combined with”

Line 58: Instead of writing out “visual inspection” here, the authors can simply say VIAA since they have already defined it.

Materials & Methods:

Line 86: What does “to women who had or not HPV diagnosis” mean? Is this referring to all women who did or did not have an HPV diagnosis which would essentially be all the women in this phase?

Lines 109-113: Were the SurveyMonkey surveys also sent to the healthcare providers or was there a different way of conducting the interviews for them?

I would probably put the sample size somewhere in the methods section. How many women did investigators want to reach and how many did they actually reach? Any power calculations to come up with a target sample size?

Clearly state the dates of data collection for phase 1 just as you've done for phase 2.

Results:

Line 168: You should define HCPs at first use.

Line 268-269: What do the authors mean when they say that the “confidence interval is 6.27”? A confidence interval by definition has a lower limit and an upper limit. Did the authors mean that 6.27 is the critical value or percent change used to calculate the confidence interval? This is unclear.

Line 270-271: Same confusion as the comment above. Are the authors meaning 90% and 95% power calculations? If your alpha level is 0.05 or 5% then essentially you would be calculating at 95% confidence interval. Please clarify.

Line 279: Why were women aged 15-20 years surveyed? General cervical cancer screening guidelines typically start in women aged 21-25 years depending on which guidelines are followed. It seems inappropriate to include women younger than 21 years as HPV prevalence is high in that age group but usually represent transient infections.

Discussion:

Are there any recommendations the authors can give to address lack of HCP training specifically in HPV-related care and cultural sensitivity for caring for this population free of biases?

Conclusions:

The authors should add a sentence or two at the end of the Conclusions section with the implications for future research or how their findings may inform future interventions. The Conclusions section should be less a restatement of results, and more a recommendation/implication of how their research strengthens future research efforts or adds to the body of literature in this field.

Author Response

Reviewer 2 :

Comments and Suggestions for Authors

It was a pleasure reading the paper “Assessing barriers encountered by women in cervical cancer screening and follow-up care in urban Bolivia, Cochabamba”. I believe it addresses an important topic and has the potential to strengthen the area of literature in barriers to HPV-related care, specifically in Bolivia. The findings have implications for future interventions that could strengthen provider training and cultural sensitivity, as well as patient knowledge and follow-up care. I have included specific recommendations below to strengthen this paper.

We thank the reviewer for its recommendations that we followed to improve our manuscript.

 Abstract:

Lines 16-17: “Timely detection of cervical cells infected with high-risk human papillomavirus (HPV) improves cervical cancer screening.” I think a clearer statement here is that timely detection improves cervical cancer outcomes or cervical cancer prevention, not screening itself.

Indeed, this is now corrected line 17 in the pdf file.

Line 17: “…actual screening coverage only reach…” should be “…actual screening coverage only reaches…”. Done. this is now corrected line 17 in the pdf file

 Introduction:

Line 48: The acronym WHO should be defined here as it is the first use. Done (line 37 in the pdf file).

Line 51: What do the authors mean by “Those”? Are you referring to HR-HPV infections? If so, I would state that to make this sentence clearer. This entire sentence can also be rephrased as it is unclear as written. This sentence was rephrased.

Line 57: “performance are not satisfactory” should be “performance is not satisfactory”. The beginning of the introduction was rewritten.

Line 58: “combined to” should be “combined with” The beginning of the introduction was rewritten.

Line 58: Instead of writing out “visual inspection” here, the authors can simply say VIAA since they have already defined it. Thank you, this has been corrected line 72 in the pdf file.

 Materials & Methods:

Line 86: What does “to women who had or not HPV diagnosis” mean? Is this referring to all women who did or did not have an HPV diagnosis which would essentially be all the women in this phase?

This is now corrected (line 163 in the pdf file).

Lines 109-113: Were the SurveyMonkey surveys also sent to the healthcare providers or was there a different way of conducting the interviews for them?

This is now more detailed in the Methods lines 228-230 in the pdf file.

I would probably put the sample size somewhere in the methods section. How many women did investigators want to reach and how many did they actually reach? Any power calculations to come up with a target sample size?

The power analysis on sample size for the phase 2 study is now given lines 191-192 and 276-277

Clearly state the dates of data collection for phase 1 just as you've done for phase 2.

 Phase 1 occurred between February 16th until July 6th 2018, as mentioned under Procedure and data collection section lines 218-219 of the pdf file.

Results:

Line 168: You should define HCPs at first use.

It is defined in the methods section line 216-217 in the pdf file.

Line 268-269: What do the authors mean when they say that the “confidence interval is 6.27”? A confidence interval by definition has a lower limit and an upper limit. Did the authors mean that 6.27 is the critical value or percent change used to calculate the confidence interval? This is unclear.

The section is moved to methods as suggested and rephrased.

Line 270-271: Same confusion as the comment above. Are the authors meaning 90% and 95% power calculations? If your alpha level is 0.05 or 5% then essentially you would be calculating at 95% confidence interval. Please clarify.

The section is moved to methods as suggested and rephrased.

Line 279: Why were women aged 15-20 years surveyed? General cervical cancer screening guidelines typically start in women aged 21-25 years depending on which guidelines are followed. It seems inappropriate to include women younger than 21 years as HPV prevalence is high in that age group but usually represent transient infections.

Indeed, in Bolivia, the law (from 3 July 2012) established that women of 18 years and more should receive a Papanicolaou test and national statistic studies on sexual and reproductive health are targeting women over 15 years. Bolivian woman are at high risk of cervical cancer, they claimed a 17 years median age at first sexual intercourse and use seldom condoms (see https://www.ine.gob.bo/index.php/publicaciones/estudio-tematico-de-salud-sexual-y-reproductiva/). So, although the previous pilot project on cervical cancer screening was performed on women with 25 years or more, here, in this survey assessing among others the level of knowledge on HPV and cervical cancer, we wanted to have a picture of population at risk , even in the future, because they are also those having to decide for their vaccination against HR-HPV or not. This explanation is now added lines  237-247 in the pdf file (sorry I already removed the track changes, so, it is not underlined here).

Discussion:

Are there any recommendations the authors can give to address lack of HCP training specifically in HPV-related care and cultural sensitivity for caring for this population free of biases?

Yes, indeed, this is now added lines 630-632, 672-673 and in the conclusions.

Conclusions:

The authors should add a sentence or two at the end of the Conclusions section with the implications for future research or how their findings may inform future interventions. The Conclusions section should be less a restatement of results, and more a recommendation/implication of how their research strengthens future research efforts or adds to the body of literature in this field. This is now done.

Round 2

Reviewer 1 Report

Thank you for addressing my comments. I really like the addition in the discussion describing the need for adequate clinical trainings for healthcare providers. The authors’ revisions have really improved the paper however, I still have remaining concerns.

1.     Please check for grammar throughout. Here are some examples:

a.     Line 53-54: Please remove mention of the low-risk HPV types. It is unclear what the authors mean by “eventually [classified] as low-risk HPV.”—HR-HPV types cannot transform into low risk types.

b.     Line 80: It is unclear what the use of “relies” is for. Please rephrase.

c.     Line 81: Please add “is” after “approach”.

d.     Line 84: Please replace “in” with “to”; and add “receive” after “…of the cases to”

e.     Line 84: Replace “patient” with “patients”.

f.      Line 85: Please replace “will be” with “is”.

g.     Line 518: Please replace “predicator” with “predictor”.

h.     Line 630: Replace “took place in the future” with “be conducted”.

i.      Line 639: Replace “believe” with “belief”.

Author Response

Thank you again for your comments as we believe the new changes improved again our manuscript.

Please check for grammar throughout. Yes, indeed, we are sorry for that. We read more carefully the text to correct grammar mistakes (highlighted in yellow in the new pdf file).

Here are some examples:

  1. Line 53-54: Please remove mention of the low-risk HPV types. It is unclear what the authors mean by “eventually [classified] as low-risk HPV.”—HR-HPV types cannot transform into low risk types. This is now removed (line 53) as it could indeed lead to confusion because of syntax error.
  2. Line 80: It is unclear what the use of “relies” is for. Please rephrase. Done line 79.
  3. Line 81: Please add “is” after “approach”. Done.
  4. Line 84: Please replace “in” with “to”; and add “receive” after “…of the cases to”. Done.
  5. Line 84: Replace “patient” with “patients”. Done.
  6. Line 85: Please replace “will be” with “is”. Done
  7. Line 518: Please replace “predicator” with “predictor”. Done.
  8. Line 630: Replace “took place in the future” with “be conducted”. Done.
  9. i. Line 639: Replace “believe” with “belief”. Done.
